# Global DNA Methylation Profiles in Peripheral Blood of WTC-Exposed Community Members with Breast Cancer

**DOI:** 10.3390/ijerph19095104

**Published:** 2022-04-22

**Authors:** Stephanie Tuminello, Yian Zhang, Lei Yang, Nedim Durmus, Matija Snuderl, Adriana Heguy, Anne Zeleniuch-Jacquotte, Yu Chen, Yongzhao Shao, Joan Reibman, Alan A. Arslan

**Affiliations:** 1Department of Population Health, New York University Langone Health, New York, NY 10016, USA; yian.zhang@nyulangone.org (Y.Z.); anne.jacquotte@nyulangone.org (A.Z.-J.); yu.chen@nyulangone.org (Y.C.); yongzhao.shao@nyulangone.org (Y.S.); 2Foundation Medicine, Cambridge, MA 02141, USA; lyang@foundationmedicine.com; 3Department of Medicine, New York University Langone Health, New York, NY 10016, USA; nedim.durmus@nyulangone.org (N.D.); joan.reibman@nyulangone.org (J.R.); 4Department of Pathology, New York University Langone Health, New York, NY 10016, USA; matija.snuderl@nyulangone.org (M.S.); adriana.heguy@nyulangone.org (A.H.); 5NYU Langone’s Genome Technology Center, New York, NY 10016, USA; 6NYU Perlmutter Comprehensive Cancer Center, New York, NY 10016, USA; 7Department of Obstetrics and Gynecology, New York University Langone Health, New York, NY 10016, USA

**Keywords:** environmental exposure, epigenome-wide association study, exposure assessment, methylation, pathway analysis, World Trade Center, 9/11, breast cancer

## Abstract

Breast cancer represents the most common cancer diagnosis among World Trade Center (WTC)-exposed community members, residents, and cleanup workers enrolled in the WTC Environmental Health Center (WTC EHC). The primary aims of this study were (1) to compare blood DNA methylation profiles of WTC-exposed community members with breast cancer and WTC-unexposed pre-diagnostic breast cancer blood samples, and (2) to compare the DNA methylation differences among the WTC EHC breast cancer cases and WTC-exposed cancer-free controls. Gene pathway enrichment analyses were further conducted. There were significant differences in DNA methylation between WTC-exposed breast cancer cases and unexposed prediagnostic breast cancer cases. The top differentially methylated genes were Intraflagellar Transport 74 (IFT74), WD repeat-containing protein 90 (WDR90), and Oncomodulin (OCM), which are commonly upregulated in tumors. Probes associated with established tumor suppressor genes (ATM, BRCA1, PALB2, and TP53) were hypermethylated among WTC-exposed breast cancer cases compared to the unexposed group. When comparing WTC EHC breast cancer cases vs. cancer-free controls, there appeared to be global hypomethylation among WTC-exposed breast cancer cases compared to exposed controls. Functional pathway analysis revealed enrichment of several gene pathways in WTC-exposed breast cancer cases including endocytosis, proteoglycans in cancer, regulation of actin cytoskeleton, axon guidance, focal adhesion, calcium signaling, cGMP-PKG signaling, mTOR, Hippo, and oxytocin signaling. The results suggest potential epigenetic links between WTC exposure and breast cancer in local community members enrolled in the WTC EHC program.

## 1. Introduction

On 11 September 2001 two hijacked airplanes, loaded with 91,000 L of jet fuel, crashed into the World Trade Center (WTC) towers [1]. Subsequent explosions and fires, burning at extreme 1800 °F temperatures, resulted in the collapse of the WTC twin towers and the adjacent building, generating large plumes of dust and smoke [1]. Initially, these plumes impacted all areas immediately adjacent to the WTC site in Manhattan, New York City (the 16 acres now known as Ground Zero) in the United States. For the following 12–18 h, winds pushed the plume east, and then to the southeast across and beyond the borough of Brooklyn [2]. Dust and smoke settled in both indoor and outdoor locations, several inches thick in some areas [2,3]. Toxic dust and smoke from the disaster took months to dissipate [1]. Thus, local community members (defined as “Survivors” under the James Zadroga 9/11 [9/11 refers to 11 September 2001] Health and Compensation Act of 2010), including local residents, local workers, cleanup workers, and students, had the potential for acute and chronic exposure to WTC-related toxic compounds. Polychlorinated chlorinated biphenyls (PCBs), polybrominated diphenol ethers (PBDEs), dioxins, furans, and polycyclic aromatic hydrocarbons (PAHs), as well as heavy metals including arsenic, beryllium, cadmium, chromium, nickel, and other elements such as copper, lead, and mercury were all measured in the WTC settled dust/smoke [2,3].

An increase in cancer incidence is now well-documented among WTC-exposed individuals, with three separate cohort studies showing that overall cancer rates of those exposed to WTC dust are 6–14% higher than expected [4,5,6]. Cancer outcomes of exposed community members, populations not involved in rescue and recovery activities, remain understudied, despite the fact that these community members experienced acute and chronic WTC dust exposure without the proper respiratory protection, and many residents and workers returned to inadequately cleaned buildings [7]. The WTC Environmental Health Center (WTC EHC) is a treatment and surveillance program for exposed community members. The WTC EHC is comprised of nearly 50% women, and is also racially and ethnically diverse, in contrast to the “Responder” cohorts [8]. Among those enrolled in the WTC EHC, breast cancer represents the most common cancer diagnosis, both overall and among women [8]. As of 31 December 2020, 13,286 individuals were enrolled and monitored in the WTC EHC, with 4635 individuals certified with a cancer diagnosis. Breast cancer was the most commonly diagnosed cancer type overall (22%) and among women (46%) [8]. The characteristics of 592 breast cancer cases (as of 31 December 2019) have recently been described [9]. Women are underrepresented in previous studies of cancer in WTC-exposed responder cohorts, hence breast cancer as a consequence of WTC exposure remains largely unexplored.

Environmental exposures to chemical components of the WTC dust, including metals, air pollutants, benzene, and organic pollutants, have all been shown to modify epigenetic status [10,11]. Thus, we have hypothesized that WTC exposure may have long-term epigenetic consequences. Our group recently reported that WTC- exposed and unexposed individuals have, on average, substantial differences in their DNA methylation profiles as measured in their peripheral blood [12]. Furthermore, although the initial pilot study was limited to cancer-free subjects, the top differentially methylated genes between WTC-exposed and unexposed community members were found to belong to several cancer-related pathways [12]. This finding suggests a potential epigenetic link between the WTC exposure and cancer development [12]. Kuan et al., also examined whether epigenetic changes in peripheral blood are associated with increased WTC exposure. Using a ranking index (ERI) they compared the DNA methylation profiles of those with low vs. high WTC exposure; however, their findings failed to reach statistical significance after multiple testing adjustments [13]. This may be due to the difficulty in quantifying WTC exposure. Moreover, previous research by Gong et al., using DNA from tissue samples, has demonstrated statistically significant differences in gene expression profiles associated with WTC exposure among prostate cancer patients, notably in genes related to immune regulation and inflammation [14].

The primary aim of this study was to compare the genome-wide DNA methylation profiles in the blood of WTC EHC women with breast cancer to a reference group of WTC-unexposed New York City residents from the NYU Women’s Health Study (NYUWHS) prospective cohort, who donated blood samples before 11 September 2001. We also aimed to characterize these changes, specifically in terms of alterations in gene pathways. As a secondary objective, we compared the DNA methylation differences between women with breast cancer and cancer-free women within the WTC EHC.

## 2. Materials and Methods

### 2.1. Study Participants

#### 2.1.1. World Trade Center Environmental Health Center (WTC EHC)

The WTC EHC is a federally designated treatment and surveillance program for WTC “Survivors” and is a “Center of Excellence” under the WTC Health Program (WTCHP). The program has been described elsewhere in detail [15,16]. Briefly, “Survivors” are defined as “persons who were present in the dust or dust cloud on 9/11 or who worked, lived, or attended school, child care centers, or adult daycare centers in the New York City disaster area” [15]. Almost 50% of Survivors experienced acute WTC dust cloud exposure on 11 September 2011, with many exposed to subsequent chronic WTC exposure from resuspended dust and fumes [15,16]. Patients self-enroll into the program and are required by law to have a “certifiable condition” such as an aerodigestive disorder or cancer [15,16].

Between March and September 2018, we invited clinic members with treatment or screening visits at the WTC EHC in Bellevue Hospital in New York City to participate in this study. Patients were asked to participate in this study and provide a venous blood sample during their routine clinical visit to the WTC EHC. In order to be eligible to participate in this study, eligible women of all races and ethnic groups enrolled in the WTC EHC had to meet the following inclusion criteria: (i) be aged 35–65 years old, (ii) given IRB-approved informed consent. Exclusion criteria included (i) being currently pregnant, (ii) currently breastfeeding, (iii) having been pregnant in the preceding 6 months prior to enrollment, (iv) having been breastfeeding in the prior 6 months prior to enrollment. Information about these patients and their WTC exposure was obtained from the WTC EHC clinical database [16]. We collected venous blood samples from 28 WTC-exposed breast cancer cases and 18 WTC-exposed cancer-free women from the WTC EHC. WTC ECH breast cancer cases and cancer-free women were frequency matched on age. The study was approved by the NYU School of Medicine and Bellevue IRB boards (IRB number: s17-01207), and all patients signed IRB-approved consent forms.

#### 2.1.2. New York University Women’s Health Study (NYUWHS)

Between March 1985 and June 1991, 14,274 women between the ages of 35 and 65 years were enrolled as volunteers in the NYUWHS at the Guttman Breast Diagnostic Institute, a mammography screening center in New York City. Eligibility was restricted to women who had not used hormonal medications, or been pregnant or lactating, in the preceding 6 months. Subjects completed a self-administered baseline questionnaire that collected information on demographic, medical, reproductive, regular physical activity, and anthropometric variables, as well as recent medication use. Cohort participation required donation of venous blood, drawn using collection tubes with anticoagulant ethylenediaminetetraacetic acid (EDTA). Blood was centrifuged, and supernatant serum and cellular precipitates were partitioned into 1-mL aliquots in capped plastic vials within 2 h after separation. Aliquots were immediately frozen at −80 °C.

As of 1 January 2021, a total of 2138 incident breast cancer cases were identified in the NYUWHS cohort after a median follow-up of 26.4 years. From these, the NYUWHS provided de-identified cell precipitate samples for DNA extraction of 24 women who developed breast cancer frequency matched by age at the time of blood donation as a reference group for the WTC EHC breast cancer cases. It is important to highlight that these samples were collected before breast cancer diagnosis, making them pre-diagnostic. However, all breast cancer diagnoses occurred before 11 September 2001. Additionally, only NYUWHS participants who developed breast cancer within 5 years of blood donation were eligible for study inclusion. The average time period between sample donation and breast cancer diagnosis for the comparison group of 24 women from the NYUWHS cohort was 3.4 years.

To address the stability of white blood cell DNA methylation profiles, we measured methylation profiles in a subset of NYUWHS subjects (*n* = 12) with samples collected between 1995 and 2018. We observed that correlation coefficients of methylation profiles cryopreserved and freshly collected samples from the same subjects were substantially higher than those between any two subjects with intraclass correlation coefficients ranging from 0.90 at CpG islands (CGI), 0.92 at CGI shore, 0.93 at gene body, and 0.95 at the promoter regions.

#### 2.1.3. Study Design

This was an observational case-control study. To meet the study’s primary objective, the epigenetic profiles of cases (WTC EHC breast cancer cases) were compared to “controls” (NYUWHS pre-diagnostic breast cancer cases). Cases and controls primarily differed by WTC exposure status. In the secondary analysis, cases (WTC EHC breast cancer cases) were compared to controls (WTC EHC cancer-free women). Here, cases and controls differed by breast cancer diagnosis. This study was designed as a pilot study.

### 2.2. Blood Sample Collection, Nucleic Acid Isolation, DNA Processing, and Differential Methylation Analysis

After obtaining informed consent, blood samples were collected during patients’ routine clinical visits to the WTC EHC. Using a standard protocol, samples were immediately centrifuged and processed to separate white blood cells (buffy coat) [17]. Additionally, reference samples from 24 pre-diagnostic breast cancer cases from the NYUWHS were identified and retrieved from storage. All (WTC EHC and NYUWHS) samples were sent to the NYU Langone’s Biospecimen Research and Development (CBRD) laboratory for DNA extraction. DNA was recovered using the PicoPure DNA extraction kit (Thermo Fisher Scientific, Boston, MA, USA), and was then subjected to bead purification with the Sphere quality control kit (Thermo Fisher Scientific, Boston, MA, USA). DNA purity and quantity were assessed using a NanoDrop spectrophotometer (NanoDrop Technologies, Wilmington, DE, USA). To improve base-pair resolution, DNA was bisulfite converted using the EZ-96 DNA methylation kit (Zymo Research, Irvine, CA, USA). The Infinium Methylation EPIC array (Illumina^®^) was used to determine the DNA methylation status of 866,562 CpG sites, following the manufacturer’s protocol.

### 2.3. Statistical Analysis and Processing of Methylation Data

All statistical analysis and modeling were performed using the open-source, statistical software R. Descriptive statistics were utilized to summarize demographic characteristics and WTC exposure by the defined groups including mean and standard deviation (SD) for continuous variables, and counts and percentages for dichotomous or categorical variables. The R package “Minfi” was used to process and analyze the methylation data [18]. Probes were quantile normalized and background adjusted, with the resulting set of samples and probes used for differentially methylated probes analysis. Using the “Finder” function, differentially methylated probes between WTC EHC cases and NYUWHS controls venous blood samples were identified. Beta-values for all 866,562 CpG probe sites tested were defined as the ratio of fluorescence intensity of the methylated probe over the overall intensity of probes. Given that this is a pilot study with a limited sample size and the number of features (CpG sites) is much larger than the sample size of study participants, Bonferroni-adjusted *p* values were computed for the tests to control the overall Type I error. This same methodology was then repeated to compare mean methylation differences between WTC EHC exposed breast cancer and cancer-free control samples. This method of global DNA methylation analysis has previously been used successfully by our group to assess epigenetic differences between WTC EHC and NYUWHS samples from healthy women in a published analysis [12]. DNA methylation probes and the information on corresponding genes were provided by Illumina^®®^. Genes associated with probes that were differentially methylated between groups were further accessed. This includes the top differentially methylated probes and corresponding genes, as well as differentially methylated probes associated with known tumor suppressor genes [19] and breast cancer-associated genes [20,21].

### 2.4. Functional Genomic Pathway Analysis

All included probes were annotated using the HumanMethylation850 manifest provided by the manufacturer (MethylationEPIC_v-1-0_B4; Illumina, San Diego, CA, USA). The University of California Santa Cruz Genome Browser database was used to obtain genomic information, including DNA sequence and coordinates of gene-coding regions [22]. All probes covering promoters and enhancers of coding genes were considered for the enrichment pathway network analysis; this rationale was adopted to limit nonspecific enrichment pathway results that may occur when all coding and noncoding genes are included. We ran in parallel Cluster profiler and the DAVID Bioinformatics Resources interrogating genes in order to detect differentially methylated genes [23,24,25]. Enrichment was determined based on a Fisher exact test value, which indicates if the overlap between genes in a cluster and a Gene Ontology term is higher than expected by chance. This pathway enrichment analysis used the high-level functional groupings provided by the Kyoto Encyclopedia of Genes and Genomes (KEGG) pathway map [26].

## 3. Results

### 3.1. WTC EHC Breast Cancer Cases vs. NYUWHS Pre-Diagnostic Breast Cancer Cases

#### 3.1.1. Characteristics of WTC EHC Breast Cancer Cases vs. NYUWHS Pre-Diagnostic Breast Cancer Cases

Characteristics of the patients who participated in the study are shown in Table 1. Compared to the NYUWHS women, WTC EHC women with breast cancer were more racially and ethnically diverse. WTC EHC breast cancer cases included a higher proportion of never smokers compared to the NYUWHS pre-diagnostic cases (61% vs. 50%, respectively), and had a higher proportion of obese women (BMI ≥ 30) (29% vs. 17%, respectively). The majority of the WTC EHC breast cancer cases reported acute WTC dust exposure on 11 September 2001 (71%). All breast cancer cases from the WTC EHC and NYUWHS cohorts in the current study were invasive ductal carcinomas.

#### 3.1.2. Methylation Profiles of WTC EHC vs. NYUWHS Pre-Diagnostic Breast Cancer Cases

To avoid the abundance of false positives due to the limited sample size, we focused on the top 5000 differentially methylated probes (Bonferroni adjusted *p* values < 0.0002) between WTC EHC vs. NYUWHS cases, the majority of which were hypermethylated in the WTC EHC group (*n* = 3092, 62%). The top differentially methylated probes were associated with the Intraflagellar Transport 74 (*IFT74),* WD repeat-containing protein 90 (*WDR90),* and Oncomodulin (*OCM)* (Table 2). Probes associated with established tumor suppressor genes were also seen to be differentially methylated between the two groups, most of which were hypermethylated among WTC-exposed breast cancer cases. These included high penetrance breast cancer genes *ATM*, *BRCA1*, *PALB2,* and *TP53*; whereas the probe associated with the *PTEN* gene was hypomethylated in the WTC EHC cases compared to the NYUWHS group (Table 3).

#### 3.1.3. Functional Pathway Enrichment Analysis for WTC EHC Breast Cancer Cases vs. NYUWHS Pre-Diagnostic Breast Cancer Cases

Several pathways were enriched for genes that appear to be differentially methylated between WTC EHC and NYUWHS samples (FDR-adjusted *p*-value < 0.01). Differential methylation of probe-associated genes belonging to several pathways was observed, including endocytosis, viral carcinogenesis, insulin resistance, phosphatidylinositol signaling system, T cell receptor signaling, and the B cell receptor signaling pathways (Figure 1).

### 3.2. WTC EHC Breast Cancer Cases vs. WTC EHC Cancer-Free Group

#### 3.2.1. Characteristics of WTC EHC Breast Cancer Cases vs. WTC-Exposed Cancer-Free Group

WTC-exposed women from WTC EHC with and without breast cancer were comparable in terms of age at blood collection, body mass index, smoking, and community member status. These groups also had similar proportions of WTC dust exposures, with each group having a high proportion of participants reporting acute exposure to the WTC dust cloud on 11 September 2001 (71% in the breast cancer group and 67% in the cancer-free group) (Table 1).

#### 3.2.2. Methylation Profiles of WTC EHC Breast Cancer Cases vs. Cancer-Free Women

Among the top 5000 differentially methylated probes most were hypomethylated in the breast cancer samples compared to the cancer-free participants (*n* = 3486, 70%). The most differentially methylated genes were *SDC2, LINCOO578,* and *PBK* (Table 4). Moreover, probes associated with tumor suppressor and breast cancer genes were also differentially methylated, notably STKLL, the promotor of which was hypermethylated among breast cancer cases. Of the 19 known tumor suppressor and breast cancer-associated genes that were differentially expressed between these two groups, 14 (74%) appeared to be hypomethylated among breast cancer samples compared to healthy controls (Table 5).

#### 3.2.3. Functional Pathway Enrichment Analysis for WTC EHC Breast Cancer Cases vs. Cancer-Free Women

Functional pathway analysis revealed differences between WTC EHC breast cancer and WTC EHC cancer-free women among several gene pathways, including endocytosis, proteoglycans in cancer, regulation of actin cytoskeleton, axon guidance, focal adhesion, calcium signaling, cGMP-PKG signaling, mTOR, Hippo, and oxytocin signaling, among others (Figure 2).

## 4. Discussion

The present study is one of the first to directly compare genome-wide DNA methylation profiles of WTC exposed (WTC EHC) breast cancer cases vs. WTC unexposed (NYUWHS) individuals who later developed breast cancer. We observed substantial differences in global DNA methylation profiles between these two groups. Specifically, there was on average increased global DNA hypermethylation among WTC EHC survivors with breast cancer compared to the unexposed Breast Cancer cases. Global DNA hypermethylation has been previously associated with environmental exposures, including from chemicals and agents comprising the WTC dust [2]. Chromium, for instance, is a toxic metal implicated in cancer, exposure to which is associated with global hypermethylation [27,28]. Arsenic has also been associated with global DNA hypermethylation, although these effects have been shown to be sex-specific, and other studies have instead observed increased global hypomethylation post-exposure [27,29]. Cadmium exposure has been associated with both global changes in DNA methylation [27,30]. In contrast, both benzene and polyaromatic carbons (PAHs) have been previously associated with global hypomethylation, although certain gene targets may still be hypermethylated [27,31]. Overall, this pattern of increased global hypermethylation associated with WTC dust exposure is consistent with our previous work comparing cancer-free WTC exposed vs. unexposed individuals [12].

The top probe-associated genes that were differentially methylated were Intraflagellar Transport 74 (IFT74), WD repeat-containing protein 90 (WDR90), and Oncomodulin (OCM), which all serve important biological functions. *IFT74* is an important component of ciliogenesis, itself an important part of the cell cycle [32]. *WDT90* is also important for cell cycle control as it is critical to centriole architecture integrity [33]. Oncomodulin (*OCM*) is a high-affinity calcium ion-binding protein found in tumors [34]. Some of the observed top differentially methylated genes have also been previously implicated in breast cancer. *TBC1D24* has been shown to play an important role in the proliferation, migration, and invasion of breast cancer cells [35]. Likewise, *FBXO11* suppression is associated with increased breast cancer cell apoptosis [36]. Moreover, the downregulation of *NCKIPSD*, the promoter of which was observed to be hypermethylated among WTC survivors with breast cancer, has previously been shown to be associated with poor prognosis among breast cancer patients [37]. *NRP2* may also be a marker of poor prognosis in breast cancer [38]. Expression of *DGKA*, whose promoter was hypomethylated among survivors with breast cancer, is associated with worse outcomes across several cancer types, and is known to be important to mammary carcinoma invasiveness [39]. The promoters of several known tumor suppressor genes (e.g., *ATM, BRCA1, and PALB2*) were all hypermethylated in WTC exposed breast cancer cases compared to the unexposed group. Hypermethylation of the promoter region is usually associated with gene silencing [40], and silencing of these specific genes are key events in breast carcinogenesis [21]. Interestingly, the promoter of the *PTEN* gene, another tumor suppressor, was hypomethylated in the WTC EHC samples compared to the unexposed group. Promoter methylation of *PTEN* is a common molecular change in breast cancer [41], so these results require further validation studies. While preliminary, these results suggest that WTC exposure may be associated with altered methylation status of important breast-cancer-related genes.

Gene pathways enrichment analysis comparing the WTC-exposed and unexposed groups was also informative. We observed the potential upregulation of several immune and cancer-related pathways in WTC-exposed breast cancer cases compared to the unexposed group, including viral carcinogenesis, T cell receptor signaling, and the B cell receptor signaling pathways. Previous research by Gong et al. has demonstrated that respiratory exposure to WTC dust can induce inflammatory and immune responses [14], and our results are consistent with these findings.

We also compared the DNA methylation profiles of WTC exposed breast cancer with exposed cancer-free individuals from the WTC EHC program. Overall, we observed increased global hypomethylation among WTC-exposed cancer cases compared to exposed controls. Global hypomethylation is a known hallmark of cancer [27]. We observed significant epigenetic differences between these two groups. As expected, several of the genes associated with top differentially methylated probes between these groups have been previously implicated in breast cancer, including *SDC2* [42], *PBK* [43], *ALOX12B* [44], *DAG1* [45], *ZNF382* [46], and *FST* [47]. The promoter of *STK11,* a high penetrance breast cancer gene [21], was commonly hypermethylated in WTC-exposed breast cancer cases compared to exposed cancer-free controls. Although requiring additional validation, these preliminary results suggest the possibility of their use to screen WTC-exposed individuals for breast cancer using DNA methylation biomarkers from blood. We hope that this work will contribute to the development of novel, minimally invasive epigenetic-based methods for screening and identification of persons at risk for breast cancer among WTC survivors. Identification of specific biological pathways associated with WTC exposure in survivors with breast cancer additionally has the potential to help guide treatments that may be most effective for this group.

The use of peripheral blood for epigenetic biomarkers of WTC exposure is a novel concept and a major strength of our study. Blood-based biomarkers have the following advantages: (a) blood collection is minimally invasive, (b) DNA can be extracted from blood, can be easily stored, and is relatively stable, (c) blood is routinely collected as part of WTC ECH monitoring visits, increasing the likelihood of patient recruitment, and (d) because blood collection is noninvasive and routine, this increases the translational and clinical utility of biomarkers identified in blood compared to tissue. Due to the stability of certain epigenetic marks on DNA, blood represents a rich, easily accessed source of information on tumor biology [48]. Previous research has demonstrated that inhalation exposure to WTC dust induces increased inflammation and oxidative stress, and that these pathophysiological changes are associated with epigenetic modifications in the lungs in animal models [49]. We therefore anticipate being able to observe epigenetic changes in blood cells, which possibly contributed towards carcinogenic effects in organs such as the breast. However, the degree of correlation between epigenetic changes in the blood cells and breast tissues of the WTC-exposed subjects is currently unknown and would need to be addressed in future studies.

Another notable strength of this study is that it fills critical research gaps. The focus of the current study is WTC-exposed community members in the WTC EHC, which is an understudied population of ethnically diverse residents, local workers, students, and cleanup workers with cancer risks that may differ from the responder populations, that predominantly consist of white males [50]. WTC-exposed women are likewise understudied, despite the high number of breast cancer diagnoses among survivors [8]. This is the first study to examine the DNA methylation profiles of WTC exposed women with breast cancer.

The use of samples from an unexposed reference group of breast cancer cases enrolled before 11 September 2001 is another strength of this current study. Identifying the appropriate non-WTC-exposed control groups has been challenging for many WTC studies to date. For example, in one of the only other studies on DNA methylation in WTC responders, Kuan et al., listed the lack of an unexposed control group as a major limitation [13]. However, a potential limitation of our study is that for WTC EHC breast cancer cases, blood for DNA extraction and subsequent methylation analysis was collected after breast cancer development, whereas for NYUWHS cases blood was collected at enrollment (pre-diagnostic), with patients developing breast cancer during follow-up. DNA methylation alterations are stable long-term changes [10] and we assume that these changes existed prior to breast cancer development. To try and ensure that NYUWHS pre-diagnostic samples were comparable to those from the WTC EHC, we limited to only those participants that developed breast cancer within 5 years of blood donation. We would expect that most if not all of these women would have had undiagnosed pre-clinical disease. Breast cancer, specifically, can develop 3–8 years before becoming palpable at routine clinical breast examinations [51]. Because DNA methylation changes occur early on in tumorigenesis [48], we would expect cancer-driving DNA methylation alterations to already be present in these pre-diagnostic blood samples. Nevertheless, future studies should replicate this work comparing the DNA methylation profiles of WTC EHC breast cancer patients to WTC unexposed breast cancer patients with blood collected at diagnosis.

When comparing the results with our prior analysis looking at DNA methylation differences between healthy WTC EHC vs. NYUWHS women [12], we observed strong consistency in differently methylated genes and gene pathways when comparing WTC EHC and NYUWHS women. Hypermethylation of the promoters of tumor suppressor genes *ATM*, *NOTCH1*, *NUP98*, *PALB2*, *TSC2*, and *WRN* had all previously been associated with WTC exposure [12] and were again observed to be hypermethylated in WTC EHC compared to NYUWHS breast cancer patients. Moreover, in both studies, we observed differential gene methylation associated with the endocytosis, viral carcinogenesis, insulin, and the phosphatidylinositol signaling system pathways [12]. This consistency further supports the hypothesis of an epigenetic link between WTC exposure and health outcomes, specifically breast cancer. Lastly, when collecting samples, we prioritized WTC EHC breast cancer cases that were recently diagnosed, aiming to limit bias from cancer treatments affecting methylation status.

This work is a preliminary feasibility study, and as such there are other limitations that must be acknowledged. Our sample size was small and limited to women only, so we cannot comment on the DNA methylation profiles of WTC-exposed male community members. To validate the findings reported here future studies using larger samples sizes are required. Small sample size also limited our ability to take into account race/ethnicity, which could affect the study results. This is especially of concern as the proportion of Caucasian participants was comparably higher in the NYUWHS group. Future larger studies should adjust for this as well as other important potential confounders, including smoking status and BMI. There is also the potential issue of cryopreservation effects on DNA methylation, particularly for the NYUWHS comparison group. To address the stability of white blood cell DNA methylation profiles after cryopreservation, we performed a sensitivity analysis (previously described in the methods section) looking at the DNA methylation profiles of a subset of NYUWHS subjects with both cryopreserved and freshly collected samples. We found substantial consistency in DNA methylation profiles collected from the same individual regardless of the preservation method. This is consistent with a recent publication on the effects of cryopreservation on the epigenetic profiles of whole blood DNA under different temperatures and storage durations, which indicated that methylation profiles of samples stored for a prolonged period (at least 20 years) were similar to those in recently collected samples [52] and had no impact on DNA methylation of various maternally-imprinted, paternally-imprinted and other tested genes [53].

## 5. Conclusions

In conclusion, we demonstrate the feasibility of using DNA extracted from peripheral blood to compare epigenetic profiles of WTC exposed women with breast cancer to unexposed breast cancer cases, as well as WTC-exposed, cancer-free controls. We report substantial differences in methylation profiles associated with WTC exposure among breast cancer cases. The observed differential methylation of specific cancer-related and tumor suppressor genes supports a potential epigenetic mechanism between WTC exposure and breast cancer development pathways. Furthermore, our work provides preliminary data suggesting epigenetic biomarkers, measured in blood, can potentially be useful for breast cancer screening and early diagnosis among WTC EHC survivors.

## Figures and Tables

**Figure 1 ijerph-19-05104-f001:**
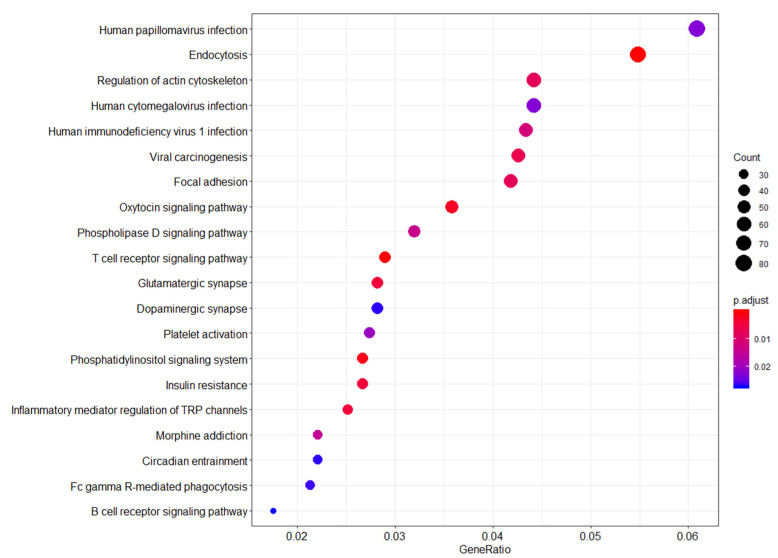
Functional genomic pathways enriched in the WTC-exposed (WTC EHC) breast cancer cases vs. unexposed (NYUWHS) breast cancer cases. **Legend:** Summary of pathway network analysis highlights the relationship between probe sets enriched in the WTC-exposed women compared to unexposed women. Y-axis shows the probe sets with significant overlap with the reference probe sets/genes from the KEGG database. X-axis shows the ratio of the number of differentially expressed probe sets/genes to the total number of genes included in the particular pathway gene set from the reference KEGG pathway database. The dot sizes are proportional to the number of overlapping probe sets/genes. The dot colors show the *p*-value adjusted for false discovery rate.

**Figure 2 ijerph-19-05104-f002:**
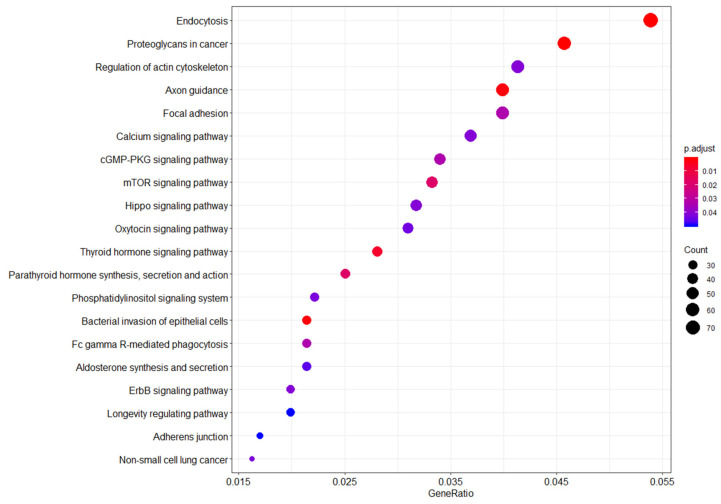
Functional genomic pathways enriched in the WTC-exposed (WTC EHC) breast cancer cases vs. WTC-exposed cancer-free women. **Legend:** Summary of pathway network analysis highlights the relationship between probe sets enriched in the WTC-exposed women compared to unexposed women. Y-axis shows the probe sets with significant overlap with the reference probe sets/genes from the KEGG database. X-axis shows the ratio of the number of differentially expressed probe sets/genes to the total number of genes included in the particular pathway gene set from the reference KEGG pathway database. The dot sizes are proportional to the number of overlapping probe sets/genes. The dot colors show the *p*-value adjusted for false discovery rate.

**Table 1 ijerph-19-05104-t001:** Descriptive characteristics of the WTC-exposed (WTC EHC) breast cancer cases, unexposed (NYUWHS) pre-diagnostic breast cancer cases, and WTC-exposed cancer-free women.

	Breast Cancer Cases, WTC EHC	Pre-Diagnostic Breast Cancer Cases, NYUWHS	Cancer-Free Controls, WTC EHC
Characteristic	*n* = 28	*n* = 24	*n* = 18
Age at blood donation, mean (SD)	60.4 (8.9)	52.1 (8.1)	57.4 (8.4)
Race/ethnicity, *n* (%)	-	-	-
Caucasian	16 (57.1)	21 (87.5)	8 (44.4)
Hispanic	4 (14.3)	1 (4.2)	8 (44.4)
African-American	7 (25.0)	1 (4.2)	1 (5.6)
Asian	1 (3.6)	1 (4.2)	1 (5.6)
Body mass index, *n* (%)	-	-	-
Normal weight (<25)	9 (32.1)	10 (41.7)	7 (39)
Overweight (25–30)	11 (39.3)	10 (41.7)	6 (33)
Obese (≥30)	8 (28.6)	4 (16.7)	5 (28)
Smoking, (*n*%)	-	-	-
Never	17 (60.7)	12 (50.0)	12 (67)
Former	10 (35.7)	11 (45.8)	5 (28)
Current	1 (3.6)	1 (4.2)	1 (5)
Community member status, *n* (%)	-	-	-
Resident	8 (28.6)	24 (100)	5 (27)
Local worker	17 (60.7)	-	11 (61)
Clean-up worker	3 (10.7)	-	2 (11)
WTC dust cloud exposure, *n* (%)			
Acute, on 9/11	20 (71.4)	-	12 (67)
Chronic, post 9/11	8 (28.6)	-	6 (33)

**Table 2 ijerph-19-05104-t002:** Top differentially methylated probes, and top 15 gene-associated probes, comparing WTC-exposed (WTC EHC) breast cancer vs. unexposed (NYUWHS) pre-diagnostic breast cancer cases.

Gene(s)	Probe ID	WTC EHC Breast Cancer Cases Methylation Value, Mean *n* = 28	NYUWHS Breast Cancer Methylation Value, Mean *n* = 24	*p* Value
IFT74	cg00877966	0.056	0.089	8.28 × 10^−17^
WDR90	cg00320059	0.785	0.737	1.49 × 10^−16^
OCM	cg20061873	0.369	0.443	3.66 × 10^−16^
NA	cg14471455	0.582	0.661	3.78 × 10^−16^
TBC1D24	cg21791024	0.681	0.613	5.78 × 10^−16^
RIN3	cg11815205	0.142	0.203	1.06 × 10^−15^
FBXO11	cg16602504	0.143	0.110	1.19 × 10^−15^
SEMA4G; MRPL43	cg19672644	0.737	0.677	1.67 × 10^−15^
GPRIN3	cg21998046	0.317	0.464	1.68 × 10^−15^
ODZ4	cg00299230	0.823	0.778	1.83 × 10^−15^
LINC00426	cg18855351	0.203	0.294	2.31 × 10^−15^
NCKIPSD	cg18849300	0.171	0.238	2.35 × 10^−15^
TIMM23B	cg09227337	0.798	0.740	2.48 × 10^−15^
ACVR1B	cg17781357	0.562	0.600	2.63 × 10^−15^
NRP2	cg01323148	0.201	0.269	2.73 × 10^−15^
DGKA	cg13634319	0.239	0.347	3.32 × 10^−15^

**Table 3 ijerph-19-05104-t003:** Methylation status of differentially expressed known tumor suppressor genes in the (WTC EHC) breast cancer vs. unexposed (NYUWHS) pre-diagnostic breast cancer cases.

Gene	Probe ID	WTC EHC Methylation Value, Mean	NYUWHS Methylation Value, Mean	Methylation Status *	Regulatory Feature	Breast Cancer Penetrance [21]
**ATM**	cg18457775	0.055	0.041	Hypermethylated	Promoter	Moderate
**BRCA1**	cg16630982	0.096	0.077	Hypermethylated	Promoter	High
**CARS**	cg25637226	0.749	0.702	Hypermethylated	NA	Poorly characterized
**CDKN2C**	cg07013994	0.052	0.040	Hypermethylated	Promoter	Poorly characterized
**CREBBP**	cg01963870	0.900	0.871	Hypermethylated	Gene	Poorly characterized
**EXT1**	cg14850625	0.066	0.045	Hypermethylated	Promoter	Poorly characterized
**NBN**	cg22881279	0.062	0.046	Hypermethylated	Promoter	Poorly characterized
**NOTCH1**	cg23457546	0.735	0.680	Hypermethylated	NA	Poorly characterized
**NUP98**	cg23511374	0.180	0.144	Hypermethylated	Promoter	Poorly characterized
**PALB2**	cg07627390	0.093	0.064	Hypermethylated	Promoter	Moderate
**PLM**	cg26861525	0.055	0.082	Hypomethylated	Promoter	Poorly characterized
**PMS1**	cg09481946	0.064	0.046	Hypermethylated	Promoter	Poorly characterized
**PTEN**	cg20849549	0.075	0.093	Hypomethylated	Promoter	High
**RUNX1**	cg21358438	0.736	0.694	Hypermethylated	NA	Poorly characterized
**TP53**	cg18198734	0.847	0.817	Hypermethylated	NA	High
**TSC2**	cg02504384	0.651	0.596	Hypermethylated	Unclassified	Poorly characterized
**WRN**	cg03410815	0.106	0.076	Hypermethylated	Promoter	Poorly characterized

* Methylation status refers to mean methylation value of the WTC exposed subjects relative to mean methylation value of unexposed subjects; NA = not available.

**Table 4 ijerph-19-05104-t004:** Top differentially methylated probes, and top 15 gene-associated probes, comparing WTC-exposed (WTC EHC) breast cancer cases vs. WTC EHC cancer-free women.

Gene(s)	Probe ID	WTC EHC Breast Cancer Cases Methylation Value, Mean *n* = 28	WTC EHC Controls Methylation Value, Mean *n* = 18	*p* Value
**SDC2**	cg10292139	0.085	0.112	2.33 × 10^−6^
**LINC00578**	cg15710610	0.573	0.653	3.93 × 10^−6^
**PBK**	cg17973773	0.697	0.630	4.11 × 10^−6^
**NA**	cg18281243	0.059	0.036	5.38 × 10^−6^
**ALOX12B**	cg11868553	0.806	0.840	6.98 × 10^−6^
**TMEM222**	cg26528255	0.523	0.555	9.14 × 10^−6^
**NA**	cg21166544	0.301	0.335	9.24 × 10^−6^
**NA**	cg25992645	0.095	0.066	9.67 × 10^−6^
**NA**	cg00187322	0.839	0.862	9.70 × 10^−6^
**ZSWIM6**	cg15221192	0.892	0.910	9.77 × 10^−6^
**DAG1**	cg11048959	0.067	0.075	1.27 × 10^−5^
**ZNF529; ZNF382**	cg19406736	0.096	0.115	1.40 × 10^−5^
**NA**	cg09562797	0.768	0.728	1.54 × 10^−5^
**NA**	cg05526438	0.623	0.650	1.56 × 10^−5^
**HIST1H4A; HIST1H3A**	cg07109238	0.055	0.062	1.59 × 10^−5^
**CNBD1**	cg20630812	0.542	0.569	1.64 × 10^−5^
**NA**	cg19366463	0.688	0.717	1.73 × 10^−5^
**MYCBPAP**	cg09079356	0.789	0.810	1.75 × 10^−5^
**NARS**	cg21911276	0.462	0.381	1.81 × 10^−5^
**ZNF607**	cg25563044	0.058	0.067	1.86 × 10^−5^
**FST**	cg22288251	0.111	0.132	1.90 × 10^−5^
**ZCCHC8**	cg02708659	0.143	0.161	1.90 × 10^−5^

**Table 5 ijerph-19-05104-t005:** Methylation status of differentially expressed known tumor suppressor genes in the WTC-exposed (WTC EHC) breast cancer cases compared to WTC EHC cancer-free women.

Gene	Probe ID	Breast Cancer Methylation Value, Mean	Cancer-Free Methylation Value, Mean	Methylation Status *	Regulatory Feature	Breast Cancer Penetrance [21]
**BCL11B**	cg23580725	0.392	0.461	Hypomethylated	NA	Poorly characterized
**CBFA2T3**	cg05540133	0.914	0.935	Hypomethylated	NA	Poorly characterized
**EXT1**	cg01483826	0.573	0.277	Hypermethylated	NA	Poorly characterized
**EXT2**	cg19330452	0.812	0.794	Hypermethylated	NA	Poorly characterized
**MAP2K4**	cg26019016	0.893	0.908	Hypomethylated	NA	Poorly characterized
**MEN1**	cg00603409	0.723	0.743	Hypomethylated	NA	Poorly characterized
**NF1**	cg07917842	0.546	0.574	Hypomethylated	NA	Poorly characterized
**NOTCH1**	cg04271687	0.054	0.059	Hypomethylated	Promoter	Poorly characterized
**NPM1**	cg17872779	0.119	0.131	Hypomethylated	Unclassified	Poorly characterized
**NR4A3**	cg13412395	0.054	0.058	Hypomethylated	NA	Poorly characterized
**NUP98**	cg20457962	0.138	0.154	Hypomethylated	Promoter	Poorly characterized
**PMS2**	cg10310847	0.095	0.114	Hypomethylated	NA	Poorly characterized
**RB1**	cg13389575	0.842	0.863	Hypomethylated	NA	Poorly characterized
**RUNX1**	cg13521940	0.137	0.110	Hypermethylated	NA	Poorly characterized
**SMARCA4**	cg17094383	0.909	0.923	Hypomethylated	NA	Poorly characterized
**STK11**	cg02671671	0.064	0.047	Hypermethylated	Promoter	High
**SUFU**	cg07636870	0.045	0.048	Hypomethylated	Promoter	Poorly characterized
**TSC2**	cg25446438	0.515	0.547	Hypomethylated	Promoter	Poorly characterized
**WT1**	cg22533573	0.129	0.111	Hypermethylated	NA	Poorly characterized

* Methylation status refers to mean methylation value of the WTC exposed subjects relative to mean methylation value of unexposed subjects; NA = not available.

## Data Availability

The data presented in this study are available on request from the corresponding author.

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
