# Peer review of "Global DNA Methylation Profiles in Peripheral Blood of WTC-Exposed Community Members with Breast Cancer"

_ijerph, 2022, doi:10.3390/ijerph19095104_

Round 1

Reviewer 1 Report

The purpose of the authors in the research entitled " Global DNA methylation profiles in peripheral blood of WTC-exposed community members with breast cancer " is very interesting.

The authors demonstrate the feasibility of using DNA extracted from peripheral blood to compare epigenetic profiles of WTC exposed women with breast cancer to unexposed breast cancer cases, as well as WTC-exposed, cancer-free controls. Additionally, the research provides preliminary data suggesting epigenetic biomarkers, measured in blood, which can potentially be useful for breast cancer screening and early diagnosis amongst WTC EHC survivors.

The hypothesis is very strong and applicable, and the results presented are well delimited. The work was well designed, making the objectives clear, and the results obtained are consistent and important.

Some minor changes should be made before publication to make the work more complete.

My suggestions are in the document (for peer review).

Reviewer 2 Report

I have several suggestions for the authors to consider.

Major

  • Study aims: (1) In the abstract, the primary aim is stated as “…to analyze blood DNA methylation profiles…”. Perhaps the aim is to “compare” blood DNA methylation profiles across the two exposure groups; “analyzing” seems to be the “how” not the “what.” (2) Similar comment for the statement of the secondary objective in Line 101 of the main text.
  • This study seems to use a case-control study design? Perhaps this should be stated in the Methods section.
  • NYUWHS control group: Were women in the NYUWHS group included if they developed breast cancer after 9/11? If so, how can you know that they were not WTC-exposed which may have led to their diagnosis? Please provide a rationale for this design.
  • For the NYUWHS group, were they frequency matched by their age when the blood sample was collected (Line 143)?
  • Given that the NYUWHS samples were drawn pre-diagnostically and the WTC EHC samples were drawn after breast cancer diagnosis, what are the implications/limitations of this?
  • Was ethnicity accounted for in this assessment? Ethnicity could influence genetic profiles including epigenetic characteristics and may bias the results if the not balanced between the comparison groups. Information on other potential confounding variables was also collected (i.e., BMI and smoking status) – was there an attempt to adjust for or to match on these variables?
  • Lines 182-183: Mentions that this is a pilot study which is the reason for the small sample size. It is suggested to mention this (i.e., that this is a pilot study) earlier in the Methods section when describing the study design.
  • There isn’t much description about the “WTC EHC cancer-free” controls – were these matched on some variables to the WTC-exposed breast cancer cases?

Minor

  • Line 99: Write out all dates (e.g., “9/11/01” could be confusing for international readers who may mistake this for November 9, 2001).
  • I realize that knowledge of 9/11 is very pervasive in the US and perhaps around the world, but I think it’s important to define the term “9/11” up front (in the Introduction) if it is going to be used throughout the text. Also, define other acronyms at first use (e.g., NYC, which is also sometimes spelled out and other times just the acronym is used, need consistency).
  • “WTC” is defined as “World Trade Center towers” (Line 44), but then sometimes the authors say, for example, “WTC twin towers” (Line 45) when only WTC (or perhaps define “WTC” to not include the “towers” term, unless that’s the intention.
  • Please specify in the Introduction that the World Trade Center and New York City is in the United States.
  • The paper should be reviewed for grammar, flow, and conciseness.

Author Response

Reviewer 2:

Major

  • Study aims: (1) In the abstract, the primary aim is stated as “…to analyze blood DNA methylation profiles…”. Perhaps the aim is to “compare” blood DNA methylation profiles across the two exposure groups; “analyzing” seems to be the “how” not the “what.” (2) Similar comment for the statement of the secondary objective in Line 101 of the main text.

As suggested these statements have been amended to “compare blood DNA methylation profiles…” in the abstract and “As a secondary objective, we compared the DNA methylation differences…” in line 102.

  • This study seems to use a case-control study design? Perhaps this should be stated in the Methods section.

This study does use a case-control study design. We have added a small subsection of the Methods section to clarify and expand on this:

2.1.3 Study Design

This was an observational case-control study. To meet the study’s primary objective, the epigenetic profiles of cases (WTC EHC breast cancer cases) were compared to “controls” (NYUWHS pre-diagnostic breast cancer cases). Cases and controls primarily differed by WTC exposure status. In the secondary analysis, cases (WTC EHC breast cancer cases) were compared to controls (WTC EHC cancer-free women). Here cases and controls differed by breast cancer diagnosis. This study was designed as a pilot study.

  • NYUWHS control group: Were women in the NYUWHS group included if they developed breast cancer after 9/11? If so, how can you know that they were not WTC-exposed which may have led to their diagnosis? Please provide a rationale for this design.

Only NYUWHS participants who developed breast cancer within 5 years of blood donation were eligible for study inclusion. All breast cancer diagnoses occurred before September 11th 2001. This is now specified in Methods section 2.1.2 lines 149-152.

  • For the NYUWHS group, were they frequency matched by their age when the blood sample was collected (Line 143)?

They were frequency matched by age at time of blood donation. This has been stated in line 147.

  • Given that the NYUWHS samples were drawn pre-diagnostically and the WTC EHC samples were drawn after breast cancer diagnosis, what are the implications/limitations of this?

This is an important limitation to our study which is now more fully addressed in the Discussion. Lines 434-443 now read:

“To try and ensure that NYUWHS pre-diagnostic samples were comparable to those from the WTC EHC, we limited to only those participants that developed breast cancer within 5 years of blood donation. We’d expect that most if not all of these women would’ve had undiagnosed pre-clinical disease. Breast cancer, specifically, can develop 3-8 years before becoming palpable at routine clinical breast examinations [52]. Because DNA methylation changes occur early on in tumorigenesis [53], we’d expect cancer-driving DNA methylation alterations to already be present in these pre-diagnostic blood samples. Nevertheless, future studies should replicate this work comparing the DNA methylation profiles of WTC EHC breast cancer patients to WTC unexposed breast cancer patients with blood collected at diagnosis.”

  • Was ethnicity accounted for in this assessment? Ethnicity could influence genetic profiles including epigenetic characteristics and may bias the results if the not balanced between the comparison groups. Information on other potential confounding variables was also collected (i.e., BMI and smoking status) – was there an attempt to adjust for or to match on these variables?

Unfortunately, due to the small sample size of this pilot study it was not possible to account for race or ethnicity in this study. This limitation is now acknowledged in the Discussion lines 461-465:

“Small sample size also limited our ability to take into account race/ethnicity, which could affect the study results. This is especially of concern as the proportion of Caucasian participants was comparably higher in the NYUWHS group. Future, larger studies should adjust for this as well as other important potential confounders, including smoking status and BMI.”

  • Lines 182-183: Mentions that this is a pilot study which is the reason for the small sample size. It is suggested to mention this (i.e., that this is a pilot study) earlier in the Methods section when describing the study design.

As suggested, this is now specified in Methods line 169: “This study was designed as a pilot study.”

  • There isn’t much description about the “WTC EHC cancer-free” controls – were these matched on some variables to the WTC-exposed breast cancer cases?

WTC ECH breast cancer cases and cancer-free women were frequency matched on age. This has been added to the Methods section lines 128-129.

Minor

  • Line 99: Write out all dates (e.g., “9/11/01” could be confusing for international readers who may mistake this for November 9, 2001).

This is a good point, thank you. It has been corrected to September 11, 2001 in line 99 and elsewhere throughout the text.

  • I realize that knowledge of 9/11 is very pervasive in the US and perhaps around the world, but I think it’s important to define the term “9/11” up front (in the Introduction) if it is going to be used throughout the text. Also, define other acronyms at first use (e.g., NYC, which is also sometimes spelled out and other times just the acronym is used, need consistency).

9/11 as a term has been defined in Introduction line 53. NYC has been spelt out as New York City throughout the text.

  • “WTC” is defined as “World Trade Center towers” (Line 44), but then sometimes the authors say, for example, “WTC twin towers” (Line 45) when only WTC (or perhaps define “WTC” to not include the “towers” term, unless that’s the intention.

As suggested, WTC has been redefied as “Word Trade Center” in line 44.

  • Please specify in the Introduction that the World Trade Center and New York City is in the United States.

This has been specified in line 48.

  • The paper should be reviewed for grammar, flow, and conciseness.

Additional edits have been made throughout to improve grammar, flow and conciseness as suggested.

Reviewer 3 Report

The study uses peripheral blood to assess DNA methylation in  World Trade Center (WTC)-exposed community members with the goal of examining breast cancer markers. The study is properly designed and the results reflect the overall data. The main weakness of the study relates to the small sample size, which the authors did address. Then again, performing methylation studies with a lot of samples is quite expensive. Can the authors elaborate a bit more on the negative vs positive control? I could not understand how those patients are different than the WTC EHC cases? If one is more diverse than the other, how can that be used as a control for it?

Author Response

Reviewer 3:

The study uses peripheral blood to assess DNA methylation in  World Trade Center (WTC)-exposed community members with the goal of examining breast cancer markers. The study is properly designed and the results reflect the overall data. The main weakness of the study relates to the small sample size, which the authors did address. Then again, performing methylation studies with a lot of samples is quite expensive. Can the authors elaborate a bit more on the negative vs positive control? I could not understand how those patients are different than the WTC EHC cases? If one is more diverse than the other, how can that be used as a control for it?

The overall objective of this study was to compare the epigenetic profiles of WTC-exposed (WTC EHC) breast cancer cases to control groups, to try and better understand the impact of WTC exposure on DNA methylation changes. Here, negative controls refer to the study participants without WTC exposure- the NYUWHS pre-diagnostic breast cancer cases. Positive controls are those with WTC exposure – WTC EHC cancer-free women. However, we acknowledge that this terminology may be confusing, and so have decided to remove the terms “negative control” and “positive control” from the manuscript.

Additionally, we have added a small subsection of the Methods section to clarify and expand on this:

2.1.3 Study Design

This was an observational case-control study. To meet the study’s primary objective, the epigenetic profiles of cases (WTC EHC breast cancer cases) were compared to “controls” (NYUWHS pre-diagnostic breast cancer cases). Cases and controls primarily differed by WTC exposure status. In the secondary analysis, cases (WTC EHC breast cancer cases) were compared to controls (WTC EHC cancer-free women). Here cases and controls differed by breast cancer diagnosis. This study was designed as a pilot study.

We acknowledge the potential for bias from having case and control groups that differ in terms of race/ethnicity. In the Discussion section lines 461-465 this limitation is now more fully addressed:

“Small sample size also limited out ability to take into account race/ethnicity, which may be biasing the results presented here. This is especially of concern as the proportion of Caucasian participants was comparably higher in the NYUWHS group. Future, larger studies should adjust for this as well as other important potential confounders, including smoking status and BMI."

Round 2

Reviewer 2 Report

The revised manuscript addressed all of my initial concerns. Just one minor comment: Please ensure that all dates are spelled out given the international audience of the journal (I found at least one instance on Line 144 where it was not: “01/01/21”).

Author Response

Thank you for finding this- all dates have now been spelt out as requested.